# Analysis of Targeted Metabolites and Molecular Structure of Starch to Understand the Effect of Glutinous Rice Paste on Kimchi Fermentation

**DOI:** 10.3390/molecules23123324

**Published:** 2018-12-14

**Authors:** Duyun Jeong, Jong-Hee Lee, Hyun-Jung Chung

**Affiliations:** 1Division of Food and Nutrition, Chonnam National University, Gwangju 61186, Korea; dersaft@naver.com; 2Research and Development Division, Advanced Process Technology and Fermentation Research Group, World Institute of Kimchi, Gwangju 61755, Korea; leejonghee@wikim.re.kr

**Keywords:** gelatinized starch, kimchi, fermentation, molecular structure, metabolites

## Abstract

*Bachu* (Chinese cabbage) kimchi, a Korean traditional fermented dish, were prepared with or without the addition of glutinous (waxy) rice paste and their characteristics including pH, total bacteria count, total starch content, sugar metabolites, and molecular structure of starch were examined periodically for 20 days to investigate the effect of adding glutinous rice paste to kimchi during fermentation. The pH and total bacteria count showed that the fermentation of kimchi added with glutinous rice paste (GRP kimchi) progressed a little more quickly than that of control kimchi without glutinous rice paste. The GRP kimchi had higher glucose content but lower fructose content than control kimchi. Interestingly, maltose was only detected in GRP kimchi during fermentation. The GRP kimchi contained much greater amount of mannitol throughout fermentation than control kimchi. Total starch content in GRP kimchi gradually decreased during fermentation, which might have contributed to its greater glucose content and the larger amount of maltose production. In GRP kimchi, peak height and area for all degrees of polymerization (DP) of starch decreased during fermentation and its average chain length decreased while the proportion of short chains increased as fermentation processed, indicating degradation of starch chains by enzymes presented in the kimchi.

## 1. Introduction

Kimchi, a Korean traditional fermented essential side dish for meals, is made through fermentation of vegetables such as Chinese cabbage (*Baechu*) and radish (*Mu*) [1]. Kimchi is classified into hundreds of varieties, depending on the types of vegetable used, region, and preparation methods [2]. Among them, *baechu* kimchi is currently the most popular and commonly used kimchi in Korea. *Baechu* kimchi is generally made by cabbage with various seasonings and additives such as red pepper powder, garlic, ginger, green onion, fermented seafood, and salts. Kimchi products are very spicy with sour, sweet, and carbonated tastes [2]. Recently, kimchi has become a globally popular food due to its health-promoting effects including anticancer, antioxidative, antidiabetic and antiobesity effects [3]. These physiological effects originate from bioactive compounds present in various ingredients in kimchi, which are amplified during fermentation process [4]. Spontaneous fermentation environment in kimchi could lead to the growth of various microorganisms [4,5]. Main microbial communities in kimchi are lactic acid bacteria (LAB) including *Lactobacillus*, *Leuconostoc*, *Weissella*, *Lactococcus*, and *Pediococcus* species [4,5,6]. Some kimchi LAB are considered probiotic strains, which are basically live microorganism that confer health-promoting benefits to hosts by improving the intestinal microbial balance [7,8,9]. Kimchi LAB are most prominent probiotics which have been receiving a tremendous attention from research field to prevent various diseases or disorders because they have been used as antibacterial, anticancer, antidiabetic, antiobesity, and antioxidant [9]. These microbial groups also contribute considerably to the safety, shelf-life, and organoleptic or nutritional properties of kimchi [10,11]. Therefore, eating kimchi is a good way to include more vegetables and probiotics in the diet to improve health [7].

The quality of kimchi mainly depends on the ingredients used, processing methods, and fermentation conditions including temperature, period, and number and type of microorganisms [12]. Moreover, proper addition of minor ingredients could produce delicious and palatable kimchi [2]. The glutinous rice paste, which is made by cooking (gelatinizing) glutinous rice flour with water, was usually added in traditional kimchi preparation in order to improve its taste and flavor, and give it viscosity [13]. Glutinous rice, also called waxy or sweet rice, is characterized mainly by its low amylose in the starch and has found extensive use in Korean traditional foods due to its sticky and soft texture after cooking [14]. There are a few existing reports regarding the effect of glutinous rice paste on kimchi fermentation [13,15,16,17]. Lee and Han [15,16] reported that the addition of glutinous rice paste to kimchi preparation could remove the off-flavor related to vegetables and improve the texture. Glutinous rice paste in kimchi can also change total acidity and reducing sugar content by affecting the growth of LAB [13,15]. Park [17] suggested that the glutinous rice paste in kimchi accelerated the fermentation process with results of pH, titratable acidity, and reducing sugar content. However, until now, a rational approach to the effect of adding glutinous rice paste during kimchi fermentation process has not been established, which makes it difficult to explain the use of glutinous rice paste in kimchi industry.

The kimchi LAB that arise from the fermentation of kimchi contributes significantly to the metabolites of kimchi products [18]. Various metabolites such as free sugars (glucose, fructose, and sucrose), organic acids (lactic and acetic acids), and other flavoring components (mannitol and amino acids) are produced during the entire kimchi fermentation process. These metabolites could be important factors for determining the taste and flavor of kimchi [18]. For determination of metabolites, high-performance liquid chromatography (HPLC)-mass spectrometry, gas chromatography (GC)-mass spectrometry, and nuclear magnetic resonance (NMR) have been used [5,18]. Since glutinous rice flour consists of about 90% starch, it may exert several effects on sugar metabolites during kimchi fermentation by its degradation to small size of glucan. However, the effects of glutinous rice paste on metabolites during kimchi fermentation and the information in degradation feature of starch in added glutinous rice have not yet been explored. Therefore, in the present study, we aimed to: (1) identify and quantify sugar-related metabolite changes in kimchi by adding glutinous rice paste during fermentation; and (2) understand structural features of starch in glutinous rice paste during fermentation. The acquired new knowledge in this study will answer the critical research question for using glutinous rice paste in kimchi industry.

## 2. Results and Discussion

### 2.1. pH of Kimchi during Fermentation

The pH over 20 days of fermentation with or without the addition of glutinous rice paste (GRP) to kimchi were monitored and results are presented in Figure 1A. The initial pH of kimchi sample with or without GRP was around 5.50. The pH of kimchi supernatant increased marginally in control kimchi or remained constant in GRP kimchi during the early stage of fermentation (0.5 day) and thereafter it substantially decreased. After 1 day of fermentation, the pH value became stable regardless of the addition of GRP and it reached approximately 3.96–4.13. Jung et al. [5] claimed that the slight increase in pH during early kimchi fermentation could be due to the released saps from various vegetables since the pH of salted cabbage (5.1) was much lower than that of seasoning (5.6) including various vegetables as reported by Yun et al. [19]. The rapid decrease in pH could be attributed to the formation of organic acids from the degradation of sugars by LABs during fermentation [5,12,18]. Similar pH profile in kimchi fermentation has been reported by Jung et al. [5] and Lee and Han [15]. During 20 days of fermentation, pH values of GRP kimchi (4.24 and 4.13 after 7 and 20 days, respectively) were slightly lower than those of control kimchi (4.00 and 3.96 after 7 and 20 days, respectively) (Figure 1A). This result is in accordance with that of Jang and Park [13] on leek kimchi and Lee and Han [15] on cabbage kimchi, who reported that the addition of glutinous rice paste to kimchi resulted in decrease in pH. Therefore, the addition of glutinous rice paste to kimchi preparation could cause rapid acceleration of the fermentation process by making LABs become predominant microorganisms due to higher amount of glucose available for producing more lactic acid, thus reducing the pH [17].

### 2.2. Count of Microorganisms during Fermentation

Total bacteria count of in kimchi samples during fermentation was highly influenced by the presence of glutinous rice paste (Figure 1B). The microbial population increased in the early stage of fermentation and then decreased. Similar result was observed by Lee and Han [16] who suggested that the rapid increase in microbial population could be due to the increased number of LABs during spontaneous fermentation since the pattern of changes between total bacterial count and lactic acid bacteria in kimchi samples during the fermentation periods was largely similar. Cho et al. [4] found that all isolates in kimchi samples belonged to a subset of lactic acid bacteria among the total of 15 species. Zabat et al. [20] also reported that the bacterial community of kimchi was dominated by species of lactic acid bacteria. The decrease in total bacteria count at late stage of fermentation could be attributed to the limitation of growth in microorganisms due to large amounts of organic acids [16]. The increase in total bacteria count inversely correlated with the decrease in pH (Figure 1). The initial count of total bacteria was higher in control kimchi than that in GRP kimchi (Figure 1B). However, after 1 day of fermentation, GRP kimchi had higher cell count than control kimchi. Final total bacteria count in GRP kimchi reached 5 × 10^5^ CFU/mL, which was lower than that of control kimchi without addition of glutinous rice paste. Similarly, Lee and Han [16] reported that even though the initial bacteria count was slightly higher in control kimchi than that in kimchi added with glutinous rice paste or wheat flour paste, the total count of bacteria and LABs of kimchi added with rice paste was much higher than that in control kimchi along with rapid decrease in pH, suggesting that the addition of glutinous rice paste or wheat flour paste could induce rapid increase in LAB population.

### 2.3. Total Starch Content in Kimchi during Fermentation

Total starch contents of kimchi decreased during fermentation over 20 days (Figure 2). Since control kimchi had no addition of glutinous rice paste containing starch, starch was not detected in its sample. The initial total starch content in GRP kimchi samples was 12.9%, which marginally changed to 12.8% during half day of fermentation. Total starch content then decreased substantially to 7.4% after 1 day of fermentation and thereafter it decreased gradually to 3.5%. Interestingly, the decrease in total starch content was positively correlated with the decrease in pH. Hahn et al. [21] reported that amylolytic enzymes including α-amylase, β-amylase, and glucoamylase contained in some basic components in kimchi (cabbage or radish) were activated during the fermentation process. The decrease in total starch content implies that activated carbohydrate enzymes might have caused the degradation of starch, leading to the release of low molecule sugars. Simultaneously, low molecular size of sugar might have been degraded to organic acid at the early stage of fermentation, resulting in a rapid drop of pH as well as increase in total bacteria count. Therefore, glutinous rice paste in kimchi could be directly converted to low molecular products by affecting pH and microorganism population, thus influencing the overall taste of kimchi. 

### 2.4. Sugars and Sugar Alcohols in Kimchi during Fermentation

Kimchi metabolites including free sugars, sugar alcohols, amino acids, and organic acids can highly influence kimchi organoleptic properties. The production of these kimchi metabolites can be significantly affected by the microorganism community during fermentation [22]. Glucose and fructose are major free sugars found in kimchi and are known to play important roles as carbon sources for LAB growth during fermentation. Levels of glucose, fructose, sucrose, and maltose during fermentation are shown in Figure 3A–D. Contents of glucose and fructose in kimchi samples rapidly decreased at the early stage of fermentation (Figure 3A,B). Similar results have been reported by several studies [10,22,23]. These decreases in glucose and fructose levels could be attributed to the consumption of free sugars as energy source for rapid growth of microorganisms and correlated well with the decrease in pH and the increase in bacterial counts. Interestingly, glucose content in GRP kimchi increased quickly at the beginning of fermentation. This increase in glucose content could be most likely due to hydrolysis of maltose liberated from starch in the glutinous rice paste. These free sugar concentrations maintained at relatively constant levels after 2 days (for glucose content) or 5 days (for fructose content) (Figure 3A,B) since the kimchi fermentation by LAB almost completed with supporting by the relatively constant pH. Interestingly, glucose content in GRP kimchi reached the minimum after 2 days and increased continuously but slowly thereafter. The glucose content in GRP kimchi reached 877 mg/100 g after 20 days of fermentation, which was twice higher than that in control kimchi (451 mg/100 g). The increase in glucose content during late fermentation period has not been reported by previous studies since glucose is used as the energy source for microorganisms. This result indicated that the production of glucose by hydrolysis of starch continually progressed, while kimchi fermentation by LAB almost completed around 5 days. On the contrary, fructose content in control kimchi was higher than that in GRP kimchi after 1 day to 20 days of fermentation. This result could be attributed to the greater consumption as carbon source of fructose in GRP kimchi, which was evidenced by its profound increase in bacterial abundance. Another possible explanation could be related to more transformation from fructose to other metabolites since it has been reported that fructose could be converted to lactate, acetate, mannitol, glycerol, or ethanol during fermentation of kimchi [10]. 

Sucrose level in kimchi without or with glutinous rice paste also substantially decreased during fermentation (Figure 3C). This decrease could be caused by its hydrolysis to glucose and fructose and the rapid uptake of LAB [24]. After 2 days of fermentation, sucrose almost hydrolyzed and its content maintained relatively constant. Sugar content in control kimchi was higher than that in GRP kimchi during the early fermentation period. However, after 2 days of fermentation, sucrose content was slightly higher in GRP kimchi than that in control kimchi. This difference could be related to diverse in environmental conditions for microorganism growth induced by added glutinous rice paste.

Major sugars detected in GRP kimchi also included maltose, which was different from previous reports [5,10] showing that maltose was not detected during fermentation (Figure 3D). Control kimchi had only 17 mg/100 g in maltose content before fermentation, but maltose was not detected after starting fermentation by complete hydrolysis to glucose. The level of maltose in GRP kimchi increased quickly at the beginning of fermentation. This increase could be attributed to hydrolysis of maltose to glucose and the production of maltose from starch by amylolytic enzymes that occurred concurrently. However, the amount of liberation of maltose from starch was much greater than its conversion to glucose. After the level of maltose reached its maximum, its content decreased rapidly and continually throughout 20 days of fermentation, a finding that correlated well with the decrease in total starch content. The decrease in maltose content could be responsible for the increase in glucose content occurring between 2 days and 20 days of fermentation in GRP kimchi. These results suggest that the added glutinous rice paste to kimchi could influence these metabolite changes with evident differences in glucose, fructose, and maltose contents during fermentation compared to control kimchi.

Sorbitol and mannitol, which influence to taste and flavor, were also detected in metabolites of kimchi (Figure 3E,F). Sorbitol was detected in a small quantity and its level marginally changed during fermentation (Figure 3E). The level of sorbitol in control kimchi was slightly higher than that in GRP kimchi. Mannitol is known as a diabetic sugar alcohol with non-cariogenic properties that imparts the cooling and refreshing taste of food [25]. Mannitol content rapidly increased during the early stage of fermentation and its content maintained relatively constant after 5 days of fermentation, which was correlated inversely with decrease in fructose content (Figure 3F). Grobben et al. [26] suggested that *Leuconostoc mesenteroids* could produce mannitol by consuming fructose with mannitol dehydrogenase. Jeong et al. [10] also reported that two *Leuconostoc* strains were identified as mannitol-producing bacteria. Interestingly, in the present study, GRP kimchi had much higher mannitol level during fermentation than control kimchi, a finding that was supported by its greater decrease in fructose content. Since the production of mannitol in kimchi has been considered as a positive feature, the addition of glutinous rice paste to kimchi could be suggested to producers.

### 2.5. Structural Analysis of Starch in Kimchi during Fermentation

Starch is a main component of the added glutinous rice paste to kimchi and it mainly consists of essentially linear amylose and highly branched amylopectin with a degree of polymerization (DP) ranging from 3 × 10^5^ to 3 × 10^6^ [27]. Amylopectin is the major fractions, account to more than 98% in waxy rice [28]. Thus, structural changes of starch in kimchi with glutinous (waxy) rice paste during fermentation were determined by chain length distribution of amylopectin. HPAEC-PAD chromatograms of starch are shown in Figure 4A. HPAEC system used in this study could detect glucans with DP up to 83. Groups with DP 6–30 showed higher normalized peak area in chromatograms, meaning major DPs in amylopectin of starch. HPAEC-PAD chromatogram could be grouped into the following chain types with applicable DP as A chains (DP 6–12), B1 chains (DP 13–24), B2 chains (DP 25–36), and B3+ chains (DP ≥ 37) [29]. The levels of amylopectin unit chains determined by the area ratio between starch from fermented kimchi and intact rice starch are grouped into different chain types and results are shown in Figure 4B. Interestingly, the chain types substantially decreased during fermentation period, suggesting that, during fermentation of kimchi, the added starch was metabolized substantially with evident decrease in total starch content (Figure 2). This result seems to reflect multiple attacks by the action of mixed enzymes in relation to the degradation of amylopectin. Similarly, Hahn et al. [21] reported that amylolytic enzymes (α-amylase, β-amylase, and glucoamylase), which could degrade α(1→4)- and α(1→6)-starch chains, were activated during kimchi fermentation. Hu et al. [30] suggested that LAB could metabolize isomalto-oligosaccharides with higher DP by cleaving α(1→4)- and α(1→6)-linkages.

HPAEC-PAD chromatogram was normalized by considering each peak area from total peak areas. Peak area ratios of different chain types reflecting structures of the residual amylopectin after fermentation are summarized in Table 1. The chain length distribution of the residual amylopectin highly changed during the fermentation of kimchi. Interestingly, the proportion of short chains of DP 6–12 significantly increased during fermentation, reaching around 50% after 5 days fermentation. On the contrary, the proportion of long chains (DP ≥ 37) proportionally decreased to around 2.0% with increase in fermentation period. The proportion of DP 13–24 slightly decreased after 2 days of fermentation and then it increased. The proportion of DP 25–26 remained relatively constant during 2 days of fermentation and thereafter it relatively decreased. These results indicate that amylolytic enzymes in kimchi excreted from ingredients or microorganisms during fermentation can trim chains of amylopectin in a uniform way, which is typical for amylase with multiple attack action [31]. This multiple attack action occurred by the amylolytic enzymes could contribute to decreasing the proportion of long chains and thus increasing the proportion of short chains. Short chains (DP 6–12) could induce weak points in the amylopectin structure, thus causing greater susceptibility to enzymatic attack. Amylolytic enzymes could easily degrade these short chains (DP 6–12) to predominately maltose. As evidenced by the data of amylopectin chain length distribution of residue after fermentation, multiple attack action of enzymes present in kimchi could degrade the cluster structure of amylopectin, producing greater amount of short chains and thus decreasing the average chain length.

## 3. Materials and Methods

### 3.1. Preparation of Kimchi, Fermentation, and Sampling

*Baechu* kimchi samples prepared according to a traditional manufacturing method with (GRP kimchi) and without (control kimchi) glutinous rice paste were obtained from Sun Kimchi Company (Gwangju, Korea). Briefly, Chinese cabbage (73.5%) was steeped in 10% (*w*/*v*) salt solution for 16 h. The steeped cabbage was washed with tap water three times and drained excess water. A seasoning mixture was prepared by mixing radishes (12.5%), Korean leek (0.5%), red pepper powder (3.0%), garlic (2.0%), ginger (0.5%), green onions (1.0%), anchovy sauce (0.5%), shrimp paste (1.5%), sugar (1.0%) and sea tangle base (4.0%). The seasoning mixture was added to the salted Chinese cabbage. Glutinous rice paste was prepared by dissolving glutinous rice flour (20 g, *Dongjinchal* cultivar) into hot water (70 mL) with continuous stirring. This was added uniformly into kimchi (10% *w*/*w*). These kimchi samples in sterilized zipper bags (400 g per bag) for triplicate analysis were divided into two sets. One set of kimchi samples (GRP kimchi) were uniformly added with glutinous rice paste (10% *w*/*w*), while the other set was control kimchi without adding glutinous rice paste. The kimchi samples were transported to laboratory in an icebox immediately after manufacturing and fermentation was performed at 20 °C for 24 h for active fermentation, then at 4 °C for 20 days according to the kimchi storage condition of Seo et al. [18]. Kimchi samples (3 packs) were taken on each fermentation for triplicate analysis. Kimchi sample in each pack was homogeneously blended (DA5500, Daesung Artlon, Seoul, Korea) and then filtrated through four layers of sterile coarse gauze (Daehan, Korea) to remove large particles [10,12,13]. These filtrates (kimchi soup) were centrifuged (3500 rpm for 15 min at 4 °C), and separated pellets and supernatants were then stored at −80 °C for starch structure and sugar-related metabolites analyses, respectively.

### 3.2. pH and Count of Microorganisms in Kimchi

The pH value of homogenized kimchi soup (liquid parts of kimchi) samples was determined using a pH meter (Orion 3-Star, Thermo-Fisher Scientific, Seoul, Korea). The total bacterial count was determined the 3M Petrifilm (St Paul, MN, USA) [32]. Briefly, one mL of each sample was spread on a 3M Petrifilm^TM^ aerobic count plate in duplicate and incubated at 30 °C for 48 h. Viable counts were expressed as log CFU/g.

### 3.3. Total Starch Content in Kimchi

Total starch content of the separated pellets in kimchi was determined using a total starch determination kit (Code K-TSTA, Megazyme International Ireland Ltd., Bray, Ireland), which is based on the Approved Method 76-13.01 [33]. This kit contains complex enzymes with α-amylase and amyloglucosidase to breakdown starch and subsequent colorimetric evaluation of the remaining glucose.

### 3.4. Sugars and Sugar Alcohols Analysis

Small molecule sugars (sucrose, maltose, glucose, and fructose) and sugar alcohols (sorbitol, and mannitol) in the supernatant of kimchi were analyzed by high-performance anion-exchange chromatography with pulsed amperometric detection (HPAEC-PAD) Dionex ICS-5000 system (Sunnyvale, CA, USA) equipped with an ED50 electrochemical detector including a 1.0 mm diameter gold electrode and a pH-Ag/AgCl combination reference electrode. The collected supernatant was filtered through a 0.45 μm filter and 10 μL of sample solution was injected into the systems consisting of a CarboPac PA1 analytical column (3 × 250 mm) and a CarboPac PA1 guard column (3 × 50 mm) with mobile phase of 150 mM NaOH under isostatic mode. The flow rate was 1.0 mL/min and the column temperature was 30 °C.

### 3.5. Structural Analysis of Starch in Kimchi

Frozen pellets in kimchi included the starch in glutinous rice paste were freeze-dried at −80 °C and 5 m torr for 72 h (FD8515, Ilshin Lab Co. Ltd., Seoul, Korea). These freeze-dried powders were used for structural analysis of starch with amylopectin chain length distribution since glutinous rice starch mainly composed of amylopectin. Amylopectin branch chain length distribution of starches was analyzed by using high-performance anion exchange chromatography with pulsed amperometric detector (HPAEC-PAD) system (Dionex ICS-5000) according to the method described by You et al. [28]. The freeze-dried powder (50 mg) from precipitate of kimchi was dispersed in 2 mL of 90% dimethyl sulfoxide and boiled in water bath with stirring for 20 min. This dispersion was precipitated with ethanol (6 mL) and centrifuged at 3000 *g* for 15 min. The isolated precipitate was re-dissolved in 2 mL of sodium acetate buffer (50 mM, pH 3.5) and then 5 μL of debranching enzyme (isoamylase, E-ISAMY, Megazyme International Ireland Ltd.) was added. The solution was incubated in a water bath at 37 °C with slow stirring (100 rpm) for 20 h. The solution was filtered through 0.45 μm nylon syringe filter and injected to the HPAEC system consisting of a CarboPac PA200 column (3 × 250 mm, Dionex Corp.) and an ED50 electrochemical detector. A gradient eluent with 150 mM NaOH and 500 mM sodium acetate in 150 mM NaOH was used as the mobile phase. The flow rate was 0.5 mL/min. The levels of amylopectin unit chains released by debranching were expressed as the area ratio (%) of each unit chains between starch in glutinous rice paste during kimchi fermentation (0.5–20 days) and intact starch (0 day) in glutinous rice paste [27]. 

### 3.6. Statistical Analysis

Data are reported as means of triplicate measurements. Statistical analyses were carried out with analysis of variance (ANOVA) and Duncan’s multiple range test (*p* < 0.05) using SPSS software system (Version 12.0, SPSS Institute Inc., Cary, NC, USA).

## 4. Conclusions

The effect of adding glutinous rice paste to kimchi during fermentation on the starch structure and sugar-related metabolites was determined in this study. This study clearly showed that the addition of glutinous rice paste to kimchi resulted in the acceleration of kimchi fermentation process and produced greater amount of glucose, maltose, and mannitol by degradation of rice starch in GRP. The amylopectin structure in added glutinous rice paste substantially changed during fermentation with respect to decrease in average chain length but increase in proportion of short chains by degrading starch chains to a large extent. Our results could provide useful information of relationships between the addition of glutinous rice paste and metabolite change in kimchi fermentation, which could be very valuable for kimchi industries.

## Figures and Tables

**Figure 1 molecules-23-03324-f001:**
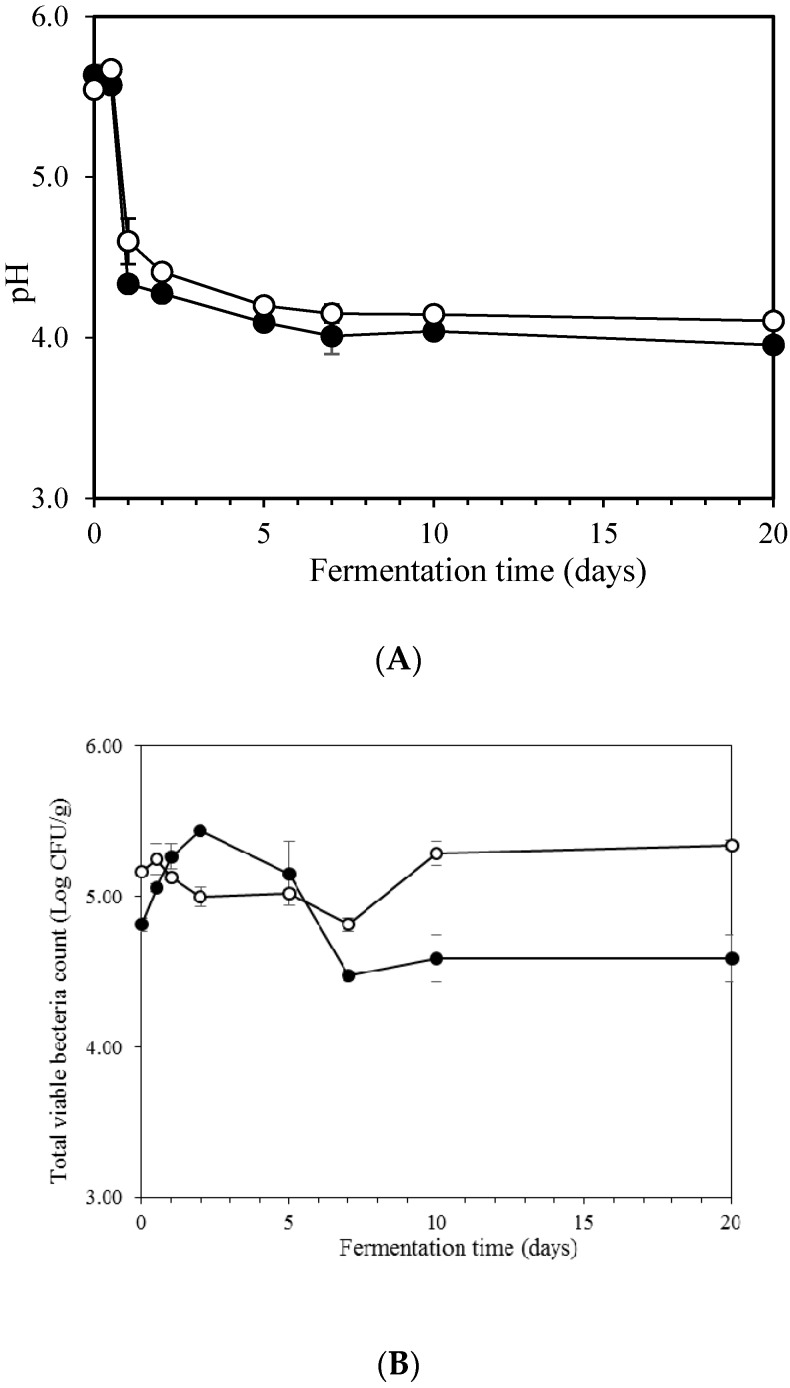
Changes in pH (**A**) and total viable bacteria count (log CFU/g) (**B**) during kimchi fermentation. ---○---, control kimchi; ---●---, GRP kimchi.

**Figure 2 molecules-23-03324-f002:**
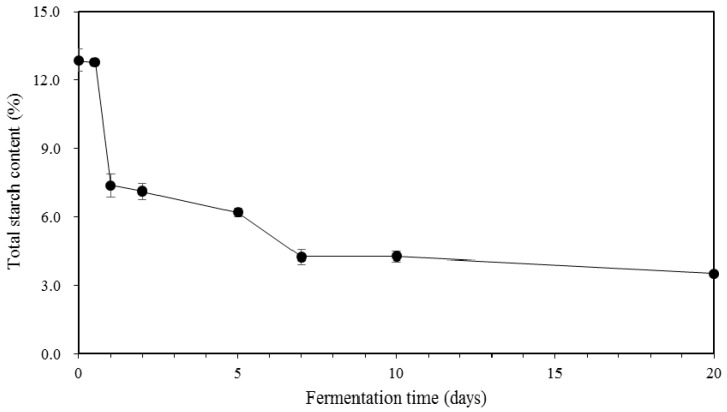
Changes in total starch content of GRP (glutinous rice paste) kimchi during fermentation.

**Figure 3 molecules-23-03324-f003:**
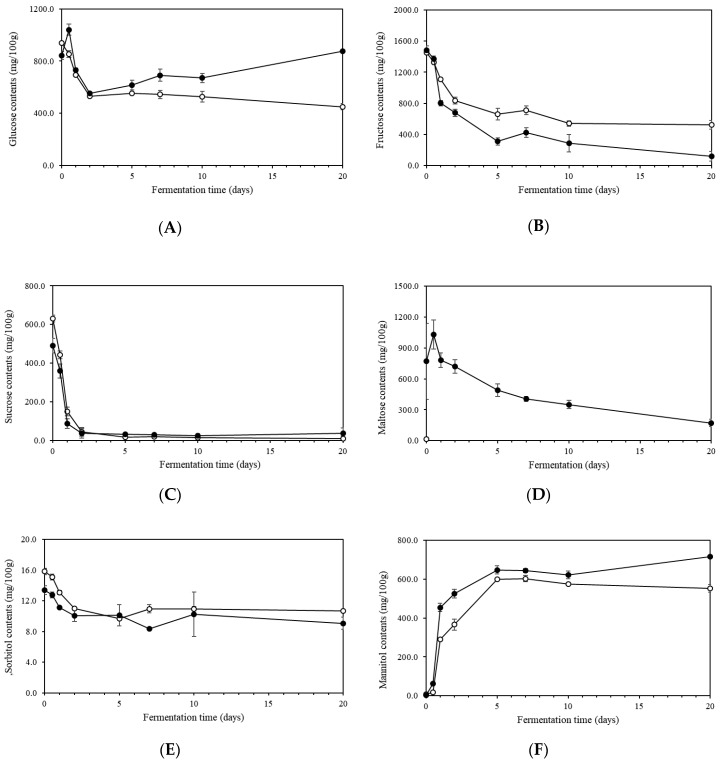
Metabolite changes in sugars (**A**–**D**) and sugar alcohols (**D**,**E**) during kimchi fermentation. (**A**), glucose; (**B**), fructose; (**C**), sucrose; (**D**), maltose; (**E**), sorbitol; (**F**), mannitol; ---○---, control kimchi; ---●---, GRP kimchi.

**Figure 4 molecules-23-03324-f004:**
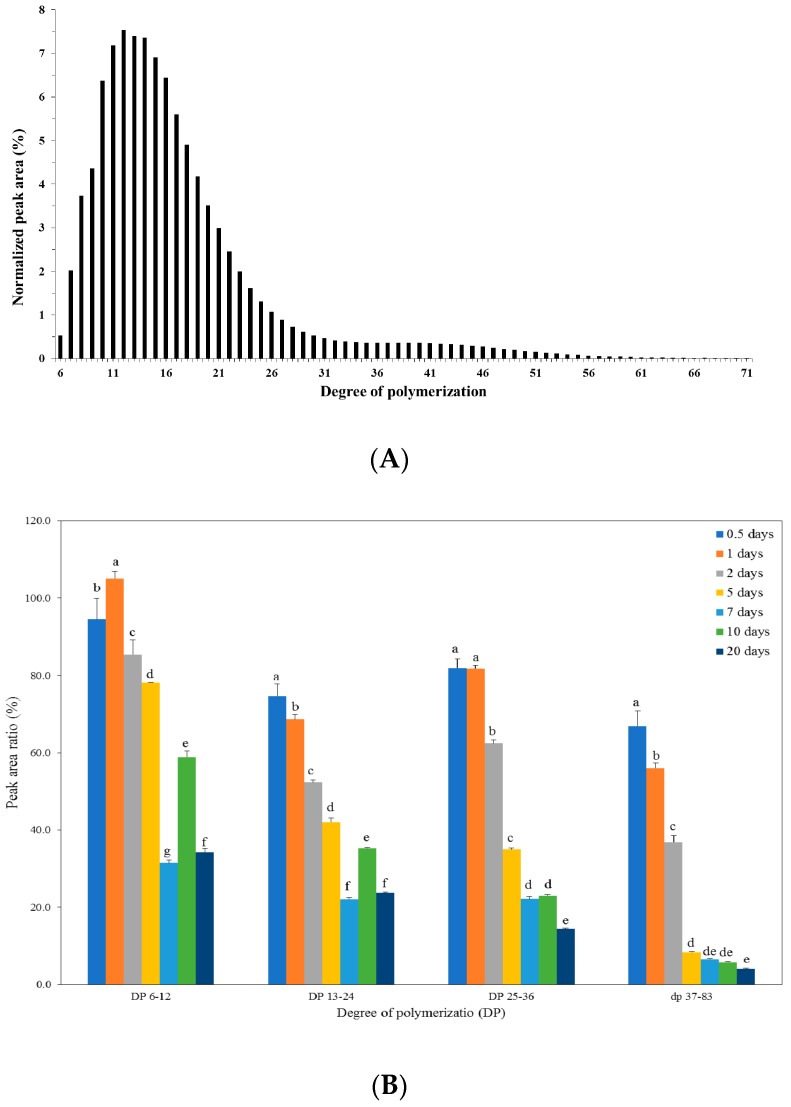
Normalized chromatograms of HPAEC (**A**) in amylopectin of glutinous rice paste before fermentation and changes of peak area ratio in each chain types (**B**) during kimchi fermentation. Different letters on bars indicate significant difference according to fermentation time (*p* < 0.05).

**Table 1 molecules-23-03324-t001:** Molecular structural changes of amylopectin in starch during kimchi fermentation.

Distribution (%)	Fermentation Time (Days)							
	0	0.5	1	2	5	7	10	20
DP 6–12	31.3 ± 0.4 ^g^	36.6 ± 0.1 ^f^	41.1 ± 0.2 ^e^	43.1 ± 0.2 ^d^	49.6 ± 0.3 ^a^	41.8 ± 0.1 ^d^	48.9 ± 0.0 ^b^	45.3 ± 0.1 ^c^
DP 13–24	46.0 ± 0.4 ^a^	42.4 ± 0.1 ^c^	39.3 ± 0.4 ^d^	38.7 ± 0.3 ^d^	39.8 ± 0.2 ^d^	43.4 ± 0.0 ^b^	42.2 ± 0.1 ^c^	45.8 ± 0.1 ^a^
DP 25–36	11.6 ± 0.1 ^b^	11.8 ± 0.1 ^ab^	11.8 ± 0.0 ^a^	11.7 ± 0.0 ^b^	8.5 ± 0.0 ^d^	11.3 ± 0.1 ^c^	6.9 ± 0.0 ^e^	7.0 ± 0.1 ^e^
DP ≥ 37	11.1 ± 0.1 ^a^	9.2 ± 0.0 ^b^	7.7 ± 0.1 ^c^	6.6 ± 0.1 ^d^	2.2 ± 0.1 ^f^	3.6 ± 0.1 ^e^	2.0 ± 0.0 ^f^	1.8 ± 0.1 ^f^

Different letters indicate significant difference according to fermentation time (*p* < 0.05).

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
