# Peer review of "Analysis of Targeted Metabolites and Molecular Structure of Starch to Understand the Effect of Glutinous Rice Paste on Kimchi Fermentation"

_molecules, 2018, doi:10.3390/molecules23123324_

Reviewer 1 Report

This manuscript gives a new insight to the investigations of traditional Korean meal- kimchi. 

I recommend the manuscript for minor review, after answering for doubts and make significant changes in some parts of manuscript, then the manuscript can be accepted to publish.

Lack of consequence in naming the products Line 13 and 31. in introduction part each sentence is a separate information and no transition line is used between them. That makes the feeling that the text is not consistent. 

Line 32 instead of "ingredients" I recommend to use "additives".

Line 64 and 65 the first part of the sentence is the same as the other just written in different words.

The information included in introduction doesn't present and describe the research which is presented in the title. And it is a little bit hardly to understand the idea. Kimchi is a meal with vegetables and different spices. Authors recommend to add glutinous rice to decrease not-positive sour taste. What is this "glutinous rice paste"? Authors mixed different names of products which makes it disordered. What I also recommend in this part to start with the description of kimchi then present what authors want to solve and then present what methods were used to achieve the goal. Also in the text appreread "kimchi soup" is this the same as presented kimchi?

No description is included for control kimchi in materials and methods part.

Is there any studies previously performed for glutinous rice?

Author Response

This manuscript gives a new insight to the investigations of traditional Korean meal- kimchi. 

I recommend the manuscript for minor review, after answering for doubts and make significant changes in some parts of manuscript, then the manuscript can be accepted to publish.

Lack of consequence in naming the products Line 13 and 31. in introduction part each sentence is a separate information and no transition line is used between them. That makes the feeling that the text is not consistent. 

Response: We are grateful for the reviewer’s constructive comments on our work. The introduction of this manuscript was extensively revised by fully following your comments.

Line 32 instead of "ingredients" I recommend to use "additives".

Response: The word was changed.

Line 64 and 65 the first part of the sentence is the same as the other just written in different words.

Response: The repeated sentence was removed.

The information included in introduction doesn't present and describe the research which is presented in the title. And it is a little bit hardly to understand the idea. Kimchi is a meal with vegetables and different spices. Authors recommend to add glutinous rice to decrease not-positive sour taste. What is this "glutinous rice paste"? Authors mixed different names of products which makes it disordered. What I also recommend in this part to start with the description of kimchi then present what authors want to solve and then present what methods were used to achieve the goal.

Response: Thanks. The introduction of this manuscript was extensively revised by fully following your comments.

Also in the text appreread "kimchi soup" is this the same as presented kimchi?

Response: Thanks for your comment. The blended kimchi sample was generally filtrated to remove large particle, which may inhibit the accurate experiment. The various references for using kimchi soup in determination of properties were added in revised manuscript.   

No description is included for control kimchi in materials and methods part.

Response: The description for control kimchi was included in the section of materials and methods.

[Baechu kimchi samples prepared according to a traditional manufacturing method with (GRP kimchi) and without (control kimchi) glutinous rice paste were obtained from Sun Kimchi Company. One set of kimchi samples (GRP kimchi) were uniformly added with glutinous rice paste (10% w/w), while the other set was control kimchi without adding glutinous rice paste.]

Is there any studies previously performed for glutinous rice?

Response: The previous studies for effect of glutinous rice paste on kimchi fermentation was introduced.

 [There are a few existing reports regarding the effect of glutinous rice paste on kimchi fermentation [13, 15, 16]. Lee and Han [15] reported that the addition of glutinous rice paste to kimchi preparation could remove the off-flavor related to vegetables and improve the texture. Glutinous rice paste in kimchi can also change total acidity and reducing sugar content by affecting the growth of LAB [13,15]. Park [16] suggested that the glutinous rice paste in kimchi accelerated the fermentation process with results of pH, titratable acidity, and reducing sugar content. However, until now, a rational approach to the effect of adding glutinous rice paste during kimchi fermentation process has not been established, which makes it difficult to explain the use of glutinous rice paste in kimchi industry.]

Reviewer 2 Report

This manuscript describes some biochemical changes during Fermentation of Kimchi material both in the absencve and presence of glutinous rice paste. This procedure appears to be of practical relevance in Food Industry. 
When reading the manuscript, however, some major concerns arose: 

The system under study is extremely complex. Two starch Sources (kimcho and Rice) have been combined and, in addition, complex populations of Bacteria are present that may vary in time. Further additions are listed in line 256-258 pg. 8. In principle, all These compounds can affect the values measured for the low molecular weight sugars/sugar derivatives.

I think it is not possible to use (and present) enzymatic debranching of polyglucan and the analysis of the resulting chains as structural features od starch.

I do not think that chains of different DP can be summarized as performed in 2.5 pg 7; Fig. 4 pg. 7; Table 1 pH 8. In HPAEC-PAD, chains of different DP are detected with different sensitivities.

It appears that only three technical replica have been used for calculating the SD values (line 310 pg. 10). Not all figures summarized as Fig. 3 exhibit SD values.

Minor points:

There seem to be errors in the text (line 23 pg. 1; line 217 pg 7; line 217 pg. 7).

Table 1 is difficult to undeerstand. What means the line starting with Average Chain Length?

In this table, the SD  quite often equals 0.

Author Response

This manuscript describes some biochemical changes during Fermentation of Kimchi material both in the absencve and presence of glutinous rice paste. This procedure appears to be of practical relevance in Food Industry. 
When reading the manuscript, however, some major concerns arose: 

The system under study is extremely complex. Two starch Sources (kimcho and Rice) have been combined and, in addition, complex populations of Bacteria are present that may vary in time. Further additions are listed in line 256-258 pg. 8. In principle, all These compounds can affect the values measured for the low molecular weight sugars/sugar derivatives.

Response: We are grateful for the reviewer’s constructive comments on our work and strongly agree with your opinions. The ingredients in kimchi does not include starch and glutinous rice consists of about 90% starch. However, as you suggested, the ingredients in kimchi definitely influence the values of the low molecular weight sugars and sugar alcohols. In our study, we would like to understand the effect of glutinous rice paste on kimchi fermentation from the comparison between kimchi with glutinous rice paste and control kimchi without glutinous rice paste since the effects of glutinous rice paste on metabolites during kimchi fermentation and the information in degradation feature of starch in added glutinous rice have not yet been explored. We have extensively revised the introduction part in manuscript to make the objective of our study clear. 

[Since glutinous rice flour consists of about 90% starch, it may many effects on sugar metabolites during kimchi fermentation by its degradation to small size of glucan. However, the effects of glutinous rice paste on metabolites during kimchi fermentation and the information in degradation feature of starch in added glutinous rice have not yet been explored. Therefore, in the present study, we aimed to: 1) identify and quantify sugar-related metabolite changes in kimchi by adding glutinous rice paste during fermentation; and 2) understand structural features of starch in glutinous rice paste during fermentation.]

I think it is not possible to use (and present) enzymatic debranching of polyglucan and the analysis of the resulting chains as structural features od starch.

Response: We are grateful for the reviewer’s constructive comments. However, the structural analysis of amylopectin chain in starch has been determined by debranching with enzymes (isoamylase) in several references.

I do not think that chains of different DP can be summarized as performed in 2.5 pg 7; Fig. 4 pg. 7; Table 1 pH 8. In HPAEC-PAD, chains of different DP are detected with different sensitivities.

Response: We are grateful for the reviewer’s constructive comments on our work. Regarding different sensitives for different DP, we strongly agree with your opinions and thus Figures 4A and 4B were revised as % of normalized peak area between starch in fermented kimchi and intact starch based on the method described by reference. The related discussion was revised.

[The level of amylopectin unit chains released by debranching for starch in glutinous rice paste during kimchi fermentation (0.5~20 days) were expressed in % of the intact starch (0 day) in glutinous rice paste [27].]

It appears that only three technical replica have been used for calculating the SD values (line 310 pg. 10). Not all figures summarized as Fig. 3 exhibit SD values.

Response: Thanks for your comments. The explanation for experimental was added in section of 3.1. Preparation of Kimchi, Fermentation, and Sampling to make it clear.

[Kimchi samples (3 packs) were taken on each fermentation for triplicate analysis. Kimchi sample in each pack was homogeneously blended (DA5500, Daesung Artlon, Seoul, Korea)]

Figure 3 was amended by considering SD values. 

Minor points:

There seem to be errors in the text (line 23 pg. 1; line 217 pg 7; line 217 pg. 7).

Response: Thanks. The errors in the text were correctly amended.

Table 1 is difficult to undeerstand. What means the line starting with Average Chain Length?

Response: Thanks for your comments. The result and related discussion for average chain length were deleted.

In this table, the SD  quite often equals 0.

Response: Thanks for your comments. The structural analysis with HPAEC-PAD was performed at least in triplicate and the reproducible results were obtained.

Reviewer 3 Report

The manuscript coded molecules-402377, entitled Analysis of Targeted Metabolites and Molecular Structure of Starch to Understand the Effect of Gelatinized Starch on Kimchi Fermentation reports intersting information about the effect of gelatinized glutinous rice on the fermentation of a traditional Korean dish, kimchi. Methods are adequate, results are statistically treated in proper way and conclusions are well-related to results.

I suggest to explain in the Abstract that kimchi is a traditional Korean dish and to highlight in the Introduction the importance of probiotics in the diet. Moreover, English needs some polishing.

Detailed observations below.

Abstract:

Line 13. After "kimchi" add "a Korean traditional fermented dish".

Line 17. Change "with glutinous rice" to "of glutinous (waxy) rice".

Line 17. Change "a little more quickly" to "faster" and add the significance (p value) in parenthesis.

Line 24. Delete "were substantially".

Line 24. Change "length was decreased" to "length decreased".

Line 25. Change "was increased" to "increased" (this mistake is frequent in the manuscript).

Introduction:

Line 51. "seasons" is unclear, reword.

Line 52. After "number" add "and type" and turn "microorganism" to plural.

Lines 52-53. Change "In addition" to "Moreover".

Line 61. Change "1) investigate identification and quantification of sugar-related" to "1) identify and quantify sugar-related".

In the Introduction add at least one sentence about the importance of probiotics in the diet, that induced a raise in the number of studies on traditional fermented foods and beverages (Pasqualone, A.; Summo, C.; Laddomada, B.; Mudura, E. and Coldea, T.E., Effect of processing variables on the physico-chemical characteristics and aroma of borş, a traditional beverage derived from wheat bran. 2018, Food Chem., 265, 242-252; Georgala A. The nutritional value of two fermented milk/cereal foods named ‘Greek Trahanas’ and ‘Turkish tarhana’: a review. J Nutr Disorders Ther. 2013; S11; Grosu-Tudor SS, Zamfir M. Probiotic potential of some lactic acid bacteria isolated from Romanian fermented vegetables. 2012, Ann RSCB.; 17: 234–239). This specification would enhance interest in the paper.

Results and discussion:

LIne 69. Delete "In this study".

Line 71. Reword "maintained" (probably you meant "reained constant"?).

Line 92. Change "due to more glucose nutrient for producing" to "due to higher amount of glucose available for producing".

Line 96. Change "Enumeration" to "Count".

Lines 116-118. Delete from "and caused the production..." to "reduction in pH" (repeats thigs already said above).

Line 120. Change "precipitate" to "decreased" and delete "were monitored".

Line 127. Change "containing some" to "contained in some".

Line 131. Change "in raid drop" to "in a rapid drop".

Line 133. Improve "transferred": could be said better with "converted", or "degraded".

Line 139. Delete "of flavors and tastes".

Line 148. Change "during the half day" to "at the beginning of".

Line 156. Amend "isused" to "is used".

Line 172. Change "difference in" to "diverse" and turn "condition" to plural.

LIne 172. After "paste" add a dot.

Line 190. Change "are contributor" to "influence".

Line 194. Delete "was" before "rapidly".

Line 195. Reword "maintained at relatively content"??

Lines 202-203. Change "favorable functionality" to "positive feature" and change "a rather positive effect" to "suggested to producers".

Line 207. Change "Amylopectin is major fractions with more than 98% of essentially amylose free waxy type" to "Amylopectin is the major fraction, accounting to more than 98% in waxy rice".

Line 212. Delete "was" before "substantially".

Lines 215-217. From "Similarly, Hahn....." to "kimchi fermentation" English is unclear (maybe a verb lacking), reword.

Line 231. Delete "was" before "highly".

Line 232. Delete "was" before "significantly".

Line 233. Delete "was" after "(DP > 37)".

LIne 235. Delete "was" before "slightly" and before "increased".

Line 236. Delete "was" after "thereafter it".

Line 247. Change "decrease" to "decreasing the".

Materials and methods:

Line 253. Possibily add a flowchart of the production process for obtaining kimchi.

Line 245. Were these kimchi purchased samples not fermented yet? Therefore, did the fermentation start at your lab, in controlled conditions? Or the fermentation was only continued in your lab? Please specify.

Line 270. Change "Enumeration" to "Count".

Author Response

The manuscript coded molecules-402377, entitled Analysis of Targeted Metabolites and Molecular Structure of Starch to Understand the Effect of Gelatinized Starch on Kimchi Fermentation reports intersting information about the effect of gelatinized glutinous rice on the fermentation of a traditional Korean dish, kimchi. Methods are adequate, results are statistically treated in proper way and conclusions are well-related to results.

I suggest to explain in the Abstract that kimchi is a traditional Korean dish and to highlight in the Introduction the importance of probiotics in the diet. Moreover, English needs some polishing.

Detailed observations below.

Response: We are grateful for the reviewer’s constructive comments on our work. The sections of abstract and introduction were revised by fully following your comments. The manuscript was revised by following your comments in detailed observations.

[Bachu (Chinese cabbage) kimchi, a Korean traditional fermented dish, were prepared with or without the addition of glutinous rice paste]

[Some kimchi LAB are considered probiotic strains, which are basically live microorganism that confer health-promoting benefits to hosts by improving the intestinal microbial balance [7-9]. Kimchi LAB are most prominent probiotics which have been receiving a tremendous attention from research field to prevent various diseases or disorders because they have been used as antibacterial, anticancer, antidiabetic, antiobesity, and antioxidant [9]. These microbial groups also contribute considerably to the safety, shelf-life, and organoleptic or nutritional properties of kimchi [10,11]. Therefore, eating kimchi is a good way to include more vegetables and probiotics in the diet to improve health [7].]

Abstract:

Line 13. After "kimchi" add "a Korean traditional fermented dish".

Response: The words were added.

Line 17. Change "with glutinous rice" to "of glutinous (waxy) rice".

Response: The word was changed

Line 17. Change "a little more quickly" to "faster" and add the significance (p value) in parenthesis.

Response: Thanks for your comments. The fermentation process could not be shown with p value. Thus, the sentence was revised.

Line 24. Delete "were substantially".

Response: The words were deleted.

Line 24. Change "length was decreased" to "length decreased".

Response: The words were changed.

Line 25. Change "was increased" to "increased" (this mistake is frequent in the manuscript).

Response: The words were changed.

Introduction:

Line 51. "seasons" is unclear, reword.

Response: Thanks for your comments. That sentence was revised by clarifying the meaning.

Line 52. After "number" add "and type" and turn "microorganism" to plural.

Response: The words were changed.

Lines 52-53. Change "In addition" to "Moreover".

Response: The word was changed

Line 61. Change "1) investigate identification and quantification of sugar-related" to "1) identify and quantify sugar-related".

Response: The words were changed.

In the Introduction add at least one sentence about the importance of probiotics in the diet, that induced a raise in the number of studies on traditional fermented foods and beverages (Pasqualone, A.; Summo, C.; Laddomada, B.; Mudura, E. and Coldea, T.E., Effect of processing variables on the physico-chemical characteristics and aroma of borş, a traditional beverage derived from wheat bran. 2018, Food Chem., 265, 242-252; Georgala A. The nutritional value of two fermented milk/cereal foods named ‘Greek Trahanas’ and ‘Turkish tarhana’: a review. J Nutr Disorders Ther. 2013; S11; Grosu-Tudor SS, Zamfir M. Probiotic potential of some lactic acid bacteria isolated from Romanian fermented vegetables. 2012, Ann RSCB.; 17: 234–239). This specification would enhance interest in the paper.

Response: Thanks for your comments.The sections of introduction were revised by fully following your comments and cited the suggested references.

[Some kimchi LAB are considered probiotic strains, which are basically live microorganism that confer health-promoting benefits to hosts by improving the intestinal microbial balance [7-9]. Kimchi LAB are most prominent probiotics which have been receiving a tremendous attention from research field to prevent various diseases or disorders because they have been used as antibacterial, anticancer, antidiabetic, antiobesity, and antioxidant [9]. These microbial groups also contribute considerably to the safety, shelf-life, and organoleptic or nutritional properties of kimchi [10,11]. Therefore, eating kimchi is a good way to include more vegetables and probiotics in the diet to improve health [7].]

Results and discussion:

LIne 69. Delete "In this study".

Response: The words were deleted.

Line 71. Reword "maintained" (probably you meant "reained constant"?).

Response: The words were changed.

Line 92. Change "due to more glucose nutrient for producing" to "due to higher amount of glucose available for producing".

Response: The words were changed.

Line 96. Change "Enumeration" to "Count".

Response: The word was changed.

Lines 116-118. Delete from "and caused the production..." to "reduction in pH" (repeats thigs already said above).

Response: Thanks. The sentences were deleted.

Line 120. Change "precipitate" to "decreased" and delete "were monitored".

Response: The sentences were revised.

Line 127. Change "containing some" to "contained in some".

Response: The words were changed.

Line 131. Change "in raid drop" to "in a rapid drop".

Response: The words were changed.

Line 133. Improve "transferred": could be said better with "converted", or "degraded".

Response: The word was changed.

Line 139. Delete "of flavors and tastes".

Response: The words were deleted.

Line 148. Change "during the half day" to "at the beginning of".

Response: The words were changed.

Line 156. Amend "isused" to "is used".

Response: The words were amended.

Line 172. Change "difference in" to "diverse" and turn "condition" to plural.

Response: The words were changed.

LIne 172. After "paste" add a dot.

Response: The dot was added.

Line 190. Change "are contributor" to "influence".

Response: The words were changed.

Line 194. Delete "was" before "rapidly".

Response: The word was deleted.

Line 195. Reword "maintained at relatively content"??

Response: The words were correctly amended.

Lines 202-203. Change "favorable functionality" to "positive feature" and change "a rather positive effect" to "suggested to producers".

Response: The words were changed.

Line 207. Change "Amylopectin is major fractions with more than 98% of essentially amylose free waxy type" to "Amylopectin is the major fraction, accounting to more than 98% in waxy rice".

Response: The sentences were revised.

Line 212. Delete "was" before "substantially".

Response: The word was deleted.

Lines 215-217. From "Similarly, Hahn....." to "kimchi fermentation" English is unclear (maybe a verb lacking), reword.

Response: Thanks for your comment. The sentences were correctly revised.

Line 231. Delete "was" before "highly".

Response: The word was deleted.

Line 232. Delete "was" before "significantly".

Response: The word was deleted.

Line 233. Delete "was" after "(DP > 37)".

Response: The word was deleted.

LIne 235. Delete "was" before "slightly" and before "increased".

Response: The word was deleted.

Line 236. Delete "was" after "thereafter it".

Response: The word was deleted.

Line 247. Change "decrease" to "decreasing the".

Response: The word was changed.

Materials and methods:

Line 253. Possibily add a flowchart of the production process for obtaining kimchi.

Response: Thanks. The brief production process of kimchi was added in revised manuscript.

[Briefly, Chinese cabbage (73.5%) was steeped in 10% (w/v) salt solution for 16 h. The steeped cabbage was washed with tap water three times and drained of excess water. A seasoning mixture was prepared by mixing radishes (12.5%), Korean leek (0.5%), red pepper powder (3.0%), garlic (2.0%), ginger (0.5%), green onions (1.0%), anchovy sauce (0.5%), shrimp paste (1.5%), sugar (1.0%) and sea tangle base (4.0%). The seasoning mixture was added to the salted Chinese cabbage.]

Line 245. Were these kimchi purchased samples not fermented yet? Therefore, did the fermentation start at your lab, in controlled conditions? Or the fermentation was only continued in your lab? Please specify.

Response: Thanks for your comment. The sentence was amended for clarifying fermentation conditions.

Line 270. Change "Enumeration" to "Count".

Response: The word was changed.

Round  2

Reviewer 2 Report

The authors did perform some changes in the manuscript that clearly increase the quality of the publication.

Still some minor changes may be done:

Line 57 pg. 2 (Introduction): Two reports exist regarding …. 

Line 73 pg. 2 (Introduction): …. starch, it may exert several effects on sugar metabolites ….

Line 176 pg 6: Legend of Fig. 3: … and sugar alcools (D, E) during kumchi …

Line 193 pg. 6: … flavor, were also detected in …

Line 310-312 pg. 9: Is that based on the starch-derived glucosyl content (see 3.3)?

Author Response

The authors did perform some changes in the manuscript that clearly increase the quality of the publication.

Still some minor changes may be done:

Line 57 pg. 2 (Introduction): Two reports exist regarding …. 

Response: Thanks for your comment. Four references for reporting the effect of glutinous rice paste on kimchi fermentation were cited in revised manuscript and thus the sentence was not changed.

Line 73 pg. 2 (Introduction): …. starch, it may exert several effects on sugar metabolites ….

Response: Thanks. The words were correctly revised.

Line 176 pg 6: Legend of Fig. 3: … and sugar alcools (D, E) during kumchi …

Response: The word was changed.

Line 193 pg. 6: … flavor, were also detected in …

Response: The word was added.

Line 310-312 pg. 9: Is that based on the starch-derived glucosyl content (see 3.3)?

Response: Thanks for your comment. The calculation method was revised to make it clear. The determination method in 3.5 is not related to that in 3.3. The total starch content was determined by the starch-derived glucosyl content after amylase treatment, whereas the amylopectin structure detected the levels of each DPs after debranching by debranching enzyme.

[The levels of amylopectin unit chains released by debranching were expressed as the area ratio (%) of each unit chains between starch in glutinous rice paste during kimchi fermentation (0.5~20 days) and intact starch (0 day) in glutinous rice paste [27].
